# Theoretical Analysis for Wireless Magnetothermal Deep Brain Stimulation Using Commercial Nanoparticles

**DOI:** 10.3390/ijms20122873

**Published:** 2019-06-12

**Authors:** Tuan-Anh Le, Minh Phu Bui, Jungwon Yoon

**Affiliations:** Gwangju Institute of Science and Technology, School of Integrated Technology, 123 Cheomdan-gwagiro, Buk-gu, Gwangju 61005, Korea; tuananhle@gist.ac.kr (T.-A.L.); buiminhphu@gist.ac.kr (M.P.B.)

**Keywords:** wireless magnetothermal stimulation, deep brain stimulation, magnetic particles

## Abstract

A wireless magnetothermal stimulation (WMS) is suggested as a fast, tetherless, and implanted device-free stimulation method using low-radio frequency (100 kHz to 1 MHz) alternating magnetic fields (AMF). As magnetic nanoparticles (MNPs) can transduce alternating magnetic fields into heat, they are targeted to a region of the brain expressing the temperature-sensitive ion channel (TRPV1). The local temperature of the targeted area is increased up to 44 °C to open the TRPV1 channels and cause an influx of Ca^2+^ sensitive promoter, which can activate individual neurons inside the brain. The WMS has initially succeeded in showing the potential of thermomagnetics for the remote control of neural cell activity with MNPs that are internally targeted to the brain. In this paper, by using the steady-state temperature rise defined by Fourier’s law, the bio-heat equation, and COMSOL Multiphysics software, we investigate most of the basic parameters such as the specific loss power (SLP) of MNPs, the injection volume of magnetic fluid, stimulation and cooling times, and cytotoxic effects at high temperatures (43–44 °C) to provide a realizable design guideline for WMS.

## 1. Introduction

Stimulation of the deep brain has shown outstanding performance for people with neurological problems such as Parkinson’s disease, essential tremor and so on [1]. Currently, the simulation method has been deployed using permanently implanted electrodes [2] and chemicals [3], acoustic [4], electromagnetic [5], or optical [6] signals. However, these methods often show un-specific stimulation or poor penetration of visible light into deep tissues, or sometimes require implanted devices for deep brain stimulation [3,6]. To overcome some of these limitations, a few groups have come up with new, competing technologies that exploit the properties of magnetic nanoparticles (MNPs). The current MNPs have shown many pre-eminent features such as a uniform size, consistent thermal, magnetic properties and coating technology [7] and are suitable for most biomedical applications such as drug delivery, imaging, and cancer hyperthermia [8,9]. MNPs (with a size range of 10–50 nm) are small enough or can be surface-modified with transferrin antibodies [10] to traverse the blood–brain barrier and can transduce alternating magnetic fields (AMF) into heat. Although MNP heating has been studied as a cell-kill therapy in magnetic hyperthermia for over 50 years [11], applying the hyperthermia method to remotely influence cellular activities has only been researched in recent years. The new method for brain stimulation using magnetic hyperthermia is called the wireless magnetothermal stimulation (WMS) method [12,13]. A method with quite a similar name, ‘transcranial magnetic stimulation’ (TMS), also exists [14]. However, it is different from WMS as it uses electromagnetic induction to stimulate the brain [14].

The WMS method has been suggested to provide a fast, tetherless, and implanted device-free method utilizing alternating magnetic fields (AMF) with ranges of 100 kHz to 1 MHz and magnetic nanoparticles (MNPs) with size range of 10–50 nm to dissipate hysteretic power loss [13]. Due to their quality of transducing alternating magnetic fields into heat, as a principle, the MNPs are targeted at neural cells expressing the temperature-sensitive ion channel (TRPV1). The local temperature of the targeted area is then increased up to 44 °C to open the TRPV1 channels and produce an influx of Ca^2+^-sensitive promoter [15,16] that can activate individual neurons inside the brain. In 2010, Huang et al., first demonstrated in [17] that the magnetic field heating of superparamagnetic nanoparticles (SNPs) could stimulate action potentials in primary hippocampal neurons. Two years later, Stanley et al., described regulated insulin production in mice with a similar method using a radio wave at 465 kHz [18]. In 2015, Chen et al. demonstrated the ability to remotely excite in vivo neuronal circuits using thermomagnetics with AMF at a frequency of 500 kHz, and a magnetic field strength of 15 kA/m [13]. In 2017, Rahul et al., used WMS for motor behavior in awake, freely moving mice [19]. Initial explorations of WMS have successfully shown the potential of thermomagnetics for the remote control of neural cell activity with MNPs that are internally targeted to the brain. However, there are no studies that have comprehensively investigated the effects of the specific loss power (SLP) of MNPs, stimulation volume (occupied by the magnetic nanoparticles), stimulation and cooling times, and cytotoxic effects at high temperatures (43–44 °C) to suggest the feasible conditions for using WMS.

To provide design guidelines for WMS experiments, this study presents a theoretical analysis of WMS using the steady-state temperature rise defined by Fourier’s law, the bio-heat equation and COMSOL Multiphysics software to investigate the basic parameters. This paper is organized as follows: in Section 2, the simulation results and discussions are presented. In Section 3, the materials and methods are introduced. Section 4 consists of the conclusion and future works.

## 2. Results and Discussion

### 2.1. Preliminary Determination of Minimum Limits for WMS without Blood Flow and Cerebrospinal Fluid (CSF)

Figure 1 shows the steady-state temperature with different SLPs, concentrations (ς_MNPs_) and injection volumes of magnetic fluid *V*_MF_ values modelled using Equation (5) and related parameters that are summarized in Table 1. The minimum values given in Table 1 are the values of SLP, ς_MNPs_ and *Q_MNPs_* for Δ*T* = 6 °C, which is the minimum requirement to conduct WMS. As can be seen, at the same value of ς_MNPs_, the SLP required for WMS decreased as the injection volume *V*_MF_ increased. Similarly, at the same value of SLP, the ς_MNPs_ required for WMS reduced as *V*_MF_ increased. Furthermore, the maximum temperature increases with the increase in *V*_MF_ if MNPs have the same SLP and ς_MNPs_ values. The minimum heating power required also reduces when *V*_MF_ is increased.

With these results, we have estimated a preliminary feasible range for WMS. Although this result is obtained without consideration of blood flow (BF) and cerebrospinal fluid (CSF), it helps us to reduce the ranges of SLP and concentration (ς_MNPs_) required for investigation (such as for *V*_MF_ = 0.6 µL, ranges required for investigation: 280 W/g ≤ SLP ≤ 1000 W/g; 28 mg/mL ≤ ς_MNPs_ ≤ 100 mg/mL), thereby significantly reducing the amount of computation required for the brain model.

### 2.2. Temperature Distribution inside the Brain before Stimulation

Before finding conditions that satisfy the temperature increase Δ*T* > 6 °C in the presence of BF and CSF, we need to check the brain model under normal conditions. A simulation without the MNP heating source is developed using the stationary study. The results presented in Figure 2 show that the temperature of brain tissue is homogeneous at 37 °C, except for a narrow region in the vicinity of its surface near the CSF area. Additionally, the temperature in the brain tissue is higher than the arterial blood temperature due to the ‘metabolic temperature shift’ *T*_m0_ = 0.36 °C [20]. The results show that this model adheres to the temperature regulation in the brain. Thus, we can use it in the next step.

### 2.3. Determination of Minimum Limits for WMS in the Presences of Blood Flow and CSF

With the effects of BF and CSF, the steady-state temperatures of MNPs are investigated again to compare them with the results of the simulation without BF and CSF given in Figure 1. The heating source *Q*_MNPs_ is situated at the center of the brain tissue (for deep brain stimulation). The temperature of MNPs is measured at the center (Figure 3) and at the edge (Figure 4) of the stimulation region. A summary of the results is shown in Table 2 The results show that in the presence of BF and CSF, the required SLP, concentration ς_MNPs_, and heating power for Δ*T* = 6 °C increase significantly by about 32%. 

These results are consistent with the fact that the blood flow plays a significant role in dissipating the heat away from the tissue with the purpose of maintaining the brain temperature at 37 °C. When the hyperthermia process is actually carried out, the blood flow reduces. However, concomitant with the reduction of blood flow, there is a proportional increase in the cerebral metabolic rate [21]. Therefore, assuming that the blood flow rate and the cerebral metabolic rate are constant throughout the WMS process is acceptable [22].

### 2.4. Prediction of a Feasbile Experiment Condition for WMS

From the results of Equation (14), only minor hyperthermic effects (*P*_C_ < 0.5%) are predicted when cells are exposed for 0.5 min to high temperatures of 43–44°C (*S* = 0.2 min^−1^) as illustrated in Figure 5. A similar result has been concluded in [13].

Since the exposure of cells to temperatures from 37–43 °C does not cause serious damage, we only consider the total time (*t*_total_) that includes heating time *t*_1_ (from 43 °C to 44 °C) and cooling time *t*_2_ (from 44 °C to 43 °C). The total time *t*_total_ should be less than 30 s, as shown in Figure 6a.

Firstly, we get a simulation time for the temperature drop from 44 °C to 43 °C in the treatment area as shown in Figure 6b (*T*_initial_ is equal to 44 °C). We see that the temperature drops below the 43 °C thermal threshold within only 3 s (or *t*_2_ < 3 s). In this investigation, we assume that the blood flow rate is constant, so the cooling time depends only on the volume of stimulation (*V*_MF_).

To keep the total time *t*_total_ below 30 s, the time *t*_1_ should be smaller than 27 s. A time-dependence study is developed with a simulation time of 25 s to find the conditions of SLP and concentration (ς_MNPs_) that sastify the *t*_1_ condition, with the *T*_initial_ of the treatment area equal to 43 °C. The final temperature results after 25 s are shown in Figure 7a–e with *V*_MF_ = 0.6 µL, 1.2 µL, 1.8 µL, 2.4 µL, and 3.0 µL, respectively. A summary of the results shown in Figure 7 is given in Table 3.

These results show that the required values of SLP, ς_MNPs_ and the heating power to carry out WMS are greater by about 36.8% than those required to satisfy only the temperature increase Δ*T* > 6 °C. These results were obtained under normal conditions. Depending on the pathology, the parameters for the simulation will need to be recalculated. Appropriate values of SLP, ς_MNPs_ and injection volume (*V*_MF_) should be selected depending on the particular application. However, it should be noted that selecting a high SLP is preferable to choosing a high concentration (ς_MNPs_) as long as the heating power is maintained. This is because with the use of high SLP, we can reduce the injection volume, thus reducing the toxic effect of magnetic particles on the body. In this study, MNPs and SNPs with a core size range of 5–31 nm are considered; magnetite, iron–platinum and maghemite are also suggested for WMS [22] as they can generate high enough heating power, are tolerable by the human body and are immuno-evasive [23]. The commercial MNPs with high stability, homogenous size, high SLP, minimal aggregation and cytotoxicity are also suitable for WMS. However, further development is still required to devise MNPs that have a small size with high SLP to increase their selectivity in targeting the stimulation area and MNPs suitable for both heating and imaging.

In addition, as technologies for measuring particle temperatures in the brain are advancing, monitoring schemes for MNP temperatures such as magnetic particle imaging (MPI) [24] should be incorporated in the WMS process. Moreover, since this simulation assumes that the MNPs are injected directly into the stimulation area, we need to consider how MNPs can reach into the stimulation area. Although a maximum of 7 mg Fe/kg or 510 mg/total dose [25] can be used in humans, to reach an MNP content that satisfies the safety condition is still a challenge. Therefore, the WMS should also be combined with a monitoring magnetic drug delivery system [26,27] and a blood–brain barrier crossing technique [28,29].

## 3. Materials and Methods

### 3.1. Fourier’s Law for Steady-State Temperature Rise

In the nanoscale world, Rabin showed that conventional heat transfer as defined by Fourier’s law is applicable for nanoparticles with sizes larger than 0.3 nm [30]. Thus, Fourier’s law is appropriate for the heat distribution analysis in biological tissues during WMS. The steady-state temperature rise at the centre of treated region is given by [30]
(1)$$\Delta T_{\mathrm{MNPs}}=\frac{Q_{\mathrm{MNPs}}\cdot d^{2}}{8\cdot k}$$
where *Q*_MNPs_ is the heat power density generated by the MNPs or volumeric heating power (W/m^3^) [31,32]; *k* is the thermal conductivity of biological tissues (*k* = 0.64 W/(m °C) [30]); and *d* is the diameter of the stimulation region.

The *Q*_MNPs_ can be determined as shown below [33]:(2)$$Q_{\mathrm{MNPs}}=\mathrm{SLP}\cdot \varsigma_{\mathrm{MNPs}}$$
where ς_MNPs_ is the concentration of MNPs, which is the ratio of the mass of MNPs and the volume of magnetic fluid (ς_MNPs_ is often given by the manufacturer), and SLP is the specific loss power to determine the amount of electromagnetic energy induced in mass unit of the MNPs. SLP can be represented by the following equation [34]:(3)$$\mathrm{SLP}=\frac{\pi \cdot \mu_{0}\cdot \chi^{\prime \prime }\left(f\right)\cdot H^{2}\cdot f}{\rho_{\mathrm{MNPs}}\cdot \phi }$$
where *f* and *H* are the magnetic field frequency and magnetic field strength, respectively, *ρ*_MNPs_ is the MNP density, *ϕ* is the MNPs volume fraction, *µ*_0_ is the permeability of free space, and *χ’’* is the imaginary part of the susceptibility.

SLP can also be explicitly related to measured heating by [35]
(4)$$\mathrm{SLP}=\frac{\rho_{\mathrm{water}}\cdot C_{\mathrm{water}}}{\varsigma_{\mathrm{MNPs}}}\cdot \frac{\Delta T}{\Delta t}$$
where *T* is the measured temperature, *t* is the time elapsed, *C*_water_ (4183 J/(kg⋅K) is the heat capacity of water, and *ρ*_water_ (998.3 kg/m^3^) is the density of water, respectively [35].

To investigate the effects of SLP of MNPs, injection volume of magnetic fluid *V*_MF_, and MNPs concentration ς_MNPs_, Equation (1) is expressed as follows:(5)ΔTMNPs=SLP⋅ςMNPs⋅6VMFπ2/38⋅k
where the injection volume of magnetic fluid is equal to *V*_MF_ = π⋅*d*^3^/6.

The magnetic field strength *H* and magnetic field frequency *f* affect the temperature rise sigificantly [31]. The commonly used range of magnetic field strength is between 0 and 50 kA/m [35,36,37,38]. The commonly used range of frequency is between 100 kHz and 700 kHz [35,36,37,38]. However, it is recommended that the product of magnetic field strength and frequency (*H f*) should not go beyond 5 × 10^9^ A/(m⋅s) [38]. To make a system with the abovementioned ranges of magnetic field strength and frequency, we can use an electromagnetic coil with soft ferromagnetic core to enhance the magnetic field and reduce the power requirement of the system [13]. However, this configuration is only advantageous for use with a small workspace, such as for testing a sample or for working at the small animal scale. For larger workspaces, this configuration requires a much larger power system compared to a solenoid coil [39,40]. A solenoid coil system, such as a heat induction system [19], is proposed for larger workspaces. However, creating a human scaled system is still a big challenge as it is required to have a huge capacity, which makes the system very expensive and raises safety issues.

In this safety condition (*H**⋅**f*
*≤* 5 × 10^9^ A/(m⋅s)), the SLP and ς_MNPs_ values for various commercial MNPs such as SHA-25 from Ocean Nanotech [38], BNF-Dextran from Micromod [36], JHU from Nano Materials Technology [36], HyperMag from NanoTherics [35] and other types of MNPs [37] are summarized in Table 4. While operating within the safety conditions, magnetic field strength and frequency values can be represented by the SLP values, and their relationship is shown in Equation (3). In addition, *Q*_MNPs_ depends on the SLP as shown in Equation (2). Thus, to simplify the analysis, we have used the SLP values instead of using both the magnetic field strength and frequency values to change *Q*_MNPs_. Utilizing values of the SLP without creating a simulation model for the magnetic field can greatly simplify the simulation computation when it is applied to the brain model.

The type of particles used in this study are MNPs or SNPs with a core size range of 5–31 nm. For particles with a size at the micro or nano-scale, the dominant source of heating is hysteresis loss [32]. In the case of SNPs, these nano-sized particles exhibit a narrow hysteresis, and the major contributor to temperature rise is heating via Néelian and Brownian relaxation [41,42].

Rabin et al. proposed that a minimal region with a diameter *d* of 0.9 mm (or an equivalent volume *V*_MF_ = 0.38 µL) occupied by nanoparticles was required to increase the temperature at the center of a tissue by 6 °C for hyperthermia with an average heating power of about 4 × 10^7^ W/m^3^ [30]. A minimum volume of MNPs is necessary to minimize their toxic effect. However, the stimulation volume can be changed depending on the area of stimulation and the type of disease. So, in this paper, we investigate the range of volume up to 3 µL, as shown in Table 1. Equation (5) does not consider the effects of blood flow (BF) and cerebrospinal fluid (CSF) as part of the main limitations. However, it is much simpler to use than the bio-heat equation described in Section 3.2. Therefore, Equation (5) is used to estimate the minimum value required for stimulation and to minimize the range of SLP, concentration ς_MNPs_, and stimulation volume *V*_MF_ required for investigation when considering the brain model.

### 3.2. Bio-Heat Transfer Model for Heat Distribution

During stimulation, unlike hyperthermia, the temperature of neural cells should be increased and then brought back to 37 °C quickly to avoid causing any thermal damage [43]. Therefore, the transient heat evolution of the infused MNPs need to be considered.

In the presence of BF and CSF, the increase of temperature is non-linear due to heat loss, and its rise rate reduces until it reaches the steady state. Therefore, the bioheat transfer model is established to describe the heat transfer in biological tissues. This model can be used to determine operation time for stimulation and cooling while taking into account the heat loss due to BF and CSF.

Although the real geometry of the human brain is complicated, we can use the simple geometry shown in Figure 8 because the temperature in the brain is (a) practically homogeneous except for a narrow region Δ in the vicinity of its surface (with only several millimeters of Δ, it is much smaller than an adult human brain size of 15 cm) and (b) practically independent of specific brain geometry [20]. The brain is modeled as a sphere of brain tissue with overlaying layers of CSF, skull and scalp. The outermost layer represents the scalp, the next layer represents the skull, the third layer represents the CSF, and the solid sphere at the center represents the brain. The model is shown in more detail in Figure 8.

The bioheat transfer equations for the brain model can be expressed as follows [22,32,35,44]:(6)$$\rho_{\mathrm{br}}C_{\mathrm{br}}\frac{\partial T_{\mathrm{br}}}{\partial t}=k_{\mathrm{br}}\nabla^{2}T_{\mathrm{br}}+\rho_{b}C_{b}\omega_{\mathrm{br}}\left(T_{b}-T_{\mathrm{br}}\right)+Q_{m\text{\_}\mathrm{br}}+Q_{\mathrm{MNPs}}$$
(7)$$\rho_{\mathrm{csf}}C_{\mathrm{csf}}\frac{\partial T_{\mathrm{csf}}}{\partial t}=k_{\mathrm{csf}}\nabla^{2}T_{\mathrm{csf}}$$
(8)$$\rho_{\mathrm{sk}}C_{\mathrm{sk}}\frac{\partial T_{\mathrm{sk}}}{\partial t}=k_{\mathrm{sk}}\nabla^{2}T_{\mathrm{sk}}$$
(9)$$\rho_{\mathrm{sc}}C_{\mathrm{sc}}\frac{\partial T_{\mathrm{sc}}}{\partial t}=k_{\mathrm{sc}}\nabla^{2}T_{\mathrm{sc}}$$
where *ρ*_br_, *ρ*_csf_, *ρ*_sk_, and *ρ*_sc_ are the densities of the brain, CSF, skull and scalp, respectively. *C*_br_, *C*_csf_, *C*_sk_, and *C*_sc_ are the specific heat of the brain, CSF, skull and scalp, respectively. *k*_br_, *k*_csf_, *k*_sk_, and *k*_sc_ are the thermal conductivity of the brain, CSF, skull and scalp, respectively. *T*_br_, *T*_csf_, *T*_sk_, *T*_sc_, *T*_b_ and *T*_env_ are the temperature of the brain, CSF, skull, scalp, arterial blood, and the surroundings of the brain, respectively. *Q*_m_br_ is the internal heat generation of the brain tissue due to cerebral metabolism. *Q*_MNPs_ is obtained using Equation (2).

The brain temperature is expected to be higher than that of the arterial blood due to the ‘metabolic temperature shift’ (*T*_m0_ = 0.36 °C) [20]. While the arterial blood temperature *T*_b_ = 36.64 °C, the initial temperature of the brain composition *T*_0_ is 37 °C. The physical parameters of the brain composition used in the brain model are given in Table 5.

In CSF, blood flow rate and metabolic heat production are absent. In the scalp and skull, these quantities are very small [45] and so can be ignored. Therefore, the temperature distribution in CSF, skull, and scalp can be described by the simplified Equations (7)–(9). We only consider the blood flow rate and metabolic rate in the brain tissue, as given in Equation (6).

Heat exchange with air can modify the uniformity of the brain temperature distribution near the brain surface. To address this issue, Equations (6)–(9) need to be solved along with the boundary conditions at the interfaces; i.e., the brain/CSF, CSF/skull, skull/scalp, and scalp/air interfaces. These boundary conditions reflect the fact that no heat dissipation takes place on the interfaces between the regions; hence, temperature and the normal component of heat flow should be continuous at each of these interfaces. The heat exchange between the scalp and air at a given temperature *T*_env_ is described by the heat transfer coefficient, *h* = 12 W/(m^2^K). Thus, we have the following boundary equations:(10)$$T_{0\text{\_}\mathrm{br}}\left(r_{\mathrm{br}}\right)=T_{0\text{\_}\mathrm{csf}}\left(r_{\mathrm{br}}\right),\;k_{\mathrm{br}}\frac{\partial T_{\mathrm{br}}}{\partial t}=k_{\mathrm{csf}}\frac{\partial T_{\mathrm{csf}}}{\partial t}$$
(11)$$T_{0\text{\_}\mathrm{csf}}\left(r_{\mathrm{csf}}\right)=T_{0\text{\_}\mathrm{sk}}\left(r_{\mathrm{csf}}\right),\;k_{\mathrm{csf}}\frac{\partial T_{\mathrm{csf}}}{\partial t}=k_{\mathrm{sk}}\frac{\partial T_{\mathrm{sk}}}{\partial t}$$
(12)$$T_{0\text{\_}\mathrm{sk}}\left(r_{\mathrm{sk}}\right)=T_{0\text{\_}\mathrm{sc}}\left(r_{\mathrm{sk}}\right),\;k_{\mathrm{sk}}\frac{\partial T_{\mathrm{sk}}}{\partial t}=k_{\mathrm{sc}}\frac{\partial T_{\mathrm{sc}}}{\partial t}$$
(13)$$k_{\mathrm{sc}}\nabla T_{\mathrm{sc}}=h\left(T_{\mathrm{sc}}-T_{\mathrm{env}}\right)$$

This model is developed using COMSOL Multiphysics software (Version 5.4) with bioheat transfer physics, stationary and time dependence Studies.

### 3.3. Cytotoxic Effects

To find a feasible condition for WMS, the probability *P*_C_ of a cell surviving an exposure for time *t* (min) to a temperature is considered using the given equation [43]:(14)$$P_{C}=1-{\left(1-{\left(1-S\right)}^{t}\right)}^{n}$$
where *S* is the inactivation rate of a molecular target of number *n*. The S-parameter varies from 0.015 to 0.2 min^−1^ in the temperature range 43–44 °C [48]. A target representing a single protein, DNA, or membrane, is inconsistent with this model. Almost all proteins present more than a few copies per cell, which is implied by the small values of *n* observed. The target number *n* is assumed to vary from 2–30 based on prior model fitting to empirical results [30]. The rate of death is highly dependent on the thermal history of the cells. Thus, the determination of the exact inactivation rate for a general condition is very complicated. Therefore, to predict a feasible condition for WMS before performing experiments, the inactivation rate is selected as the maximum value or *S* = 0.2 min^−1^.

## 4. Conclusions

Stimulation of the deep brain has shown outstanding performance for people with neurological problems. Although there are several more invasive stimulation methods, WMS has emerged as a promising, less invasive alternative with numerous advantages such as being fast, tetherless, and implanted device-free. To ensure the safe operation of WMS, the magnetic strength and frequency are limited (*H*∙*f* ≤ 5 × 10^9^ A/(m⋅s). With the current technology, only the selction of a range of SLP and ς_MNPs_ is possible, which we have done based on previous studies. Then, we presented a simulation scheme for WMS using steady-state temperature equations based on Fourier’s law to determine a preliminary requirement for SLP, concentration, MNP volume and heating power. A simple brain model satisfying the temperature regulation in the brain was developed using the bioheat transfer equation in Comsol Multiphysics^TM^ software. In the presence of BF and CSF, the requirements for SLP, concentration, MNPs volume and heating power needed were found to be higher than those found using the equations based on Fourier’s law. In addition, a feasible condition for WMS was suggested based on the cytotoxic effects and the simulation results. A preliminary review of the simulation results shows that the heating power required to carry out WMS is greater by about 36.8% than that required to satisfy only the temperature increase Δ*T* = 6 °C. This result will be quite helpful for researchers in selecting adequate MNPs to use for WMS. Our future work will include practical experiments in a mouse brain, as well as the development of a brain model that can depict real circumstances.

## Figures and Tables

**Figure 1 ijms-20-02873-f001:**
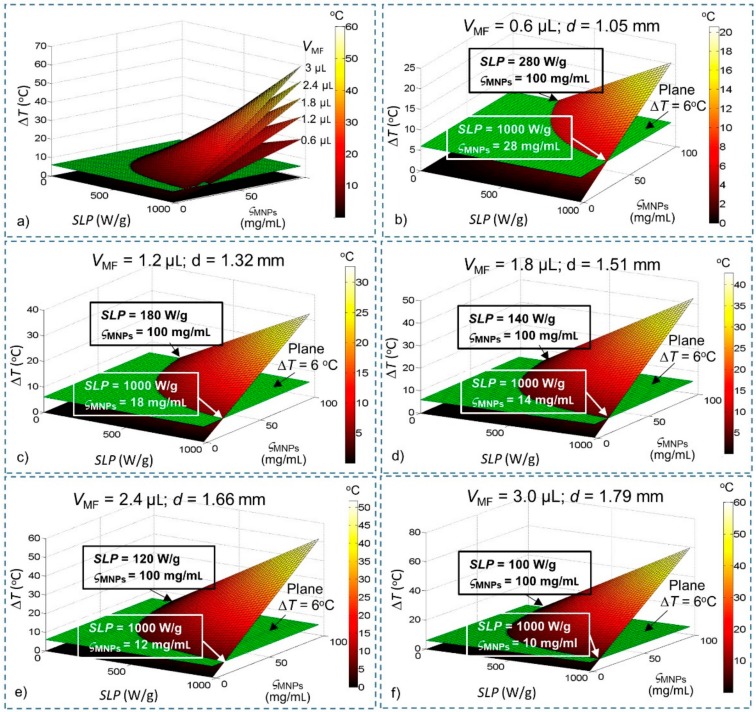
Steady-state temperature change derived from Fourier’s law as a function of specific loss power (SLP), concentration (ς_MNPs_) and volume (*V*_MF_) of magnetic nanoparticles (MNPs). Conditions that satisfy the temperature increase Δ*T* > 6 °C lie above the green plane (Δ*T* = 6 °C). (**a**) All the different values of *V*_MF_; (**b**) *V*_MF_ = 0.6 µL; (**c**) *V*_MF_ = 1.2 µL; (**d**) *V*_MF_ = 1.8 µL; (**e**) *V*_MF_ = 2.4 µL; and (**f**) *V*_MF_ = 3.0 µL.

**Figure 2 ijms-20-02873-f002:**
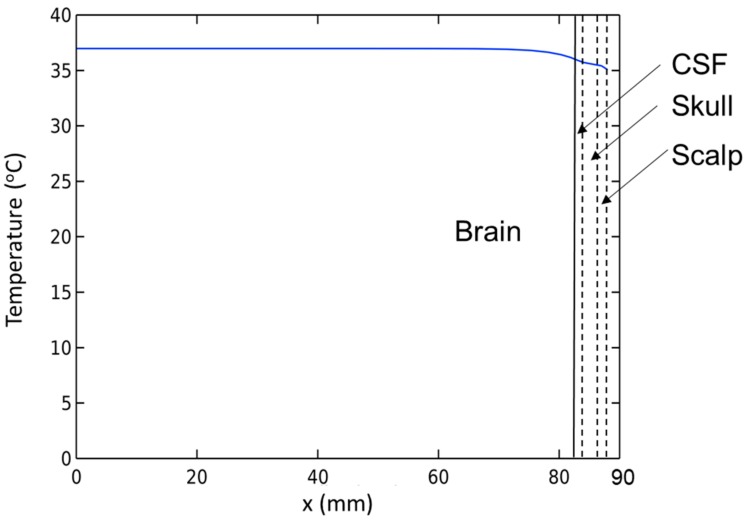
Temperature distribution inside the brain before stimulation.

**Figure 3 ijms-20-02873-f003:**
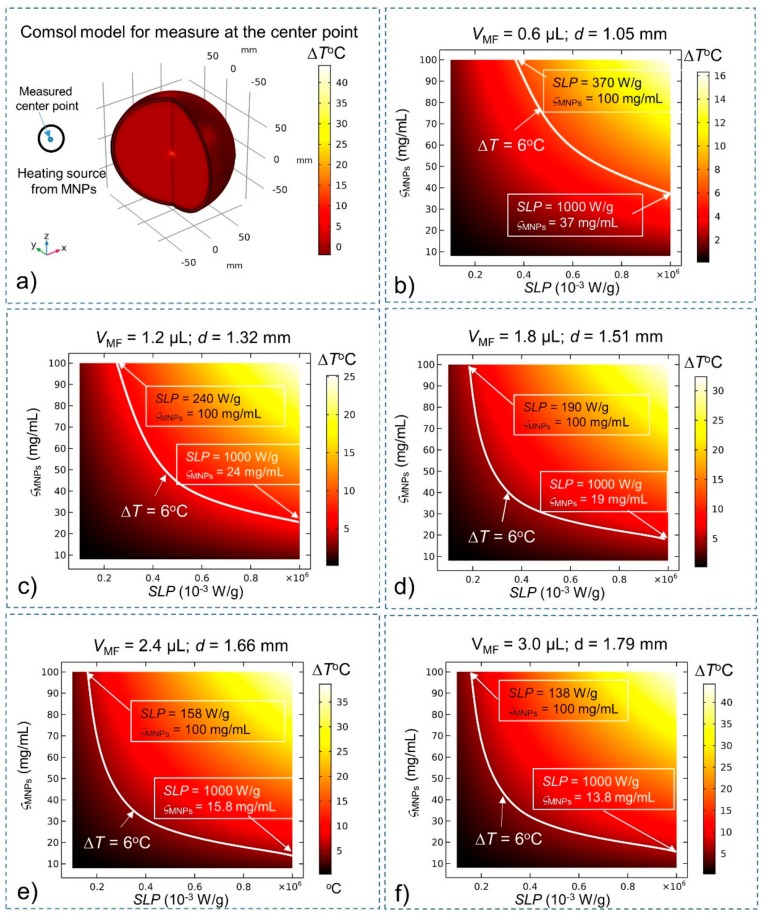
Steady-state temperature at the center of the stimulation region obtained using the bio-heat equation as a function of the SLP, concentration (ς_MNPs_) and volume (*V*_MF_) of MNPs. Conditions that satisfy the temperature increase Δ*T* > 6 °C lie above the white curve (*ΔT* = 6 °C). (**a**) Comsol Multiphysics model for measuring temperature at the center of the stimulation region; (**b**) *V*_MF_ = 0.6 µL; (**c**) *V*_MF_ = 1.2 µL; (**d**) V_MF_ = 1.8 µL; (**e**) *V*_MF_ = 2.4 µL; and (**f**) *V*_MF_ = 3.0 µL.

**Figure 4 ijms-20-02873-f004:**
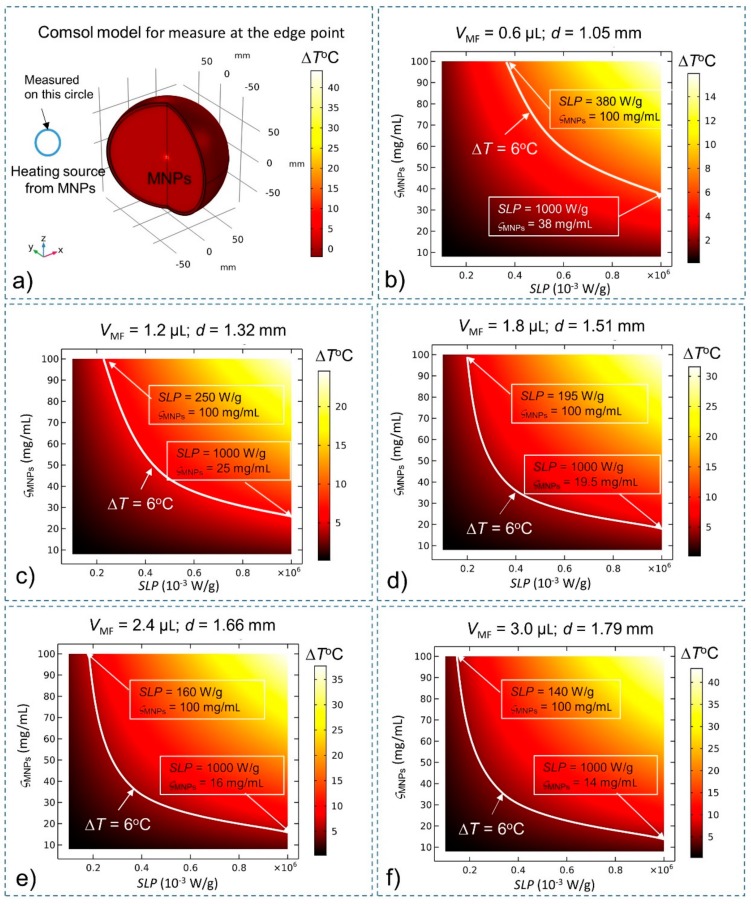
Steady-state temperature at the edge of the stimulation region while considering BF and CSF found using the bio-heat equation as a function of the SLP, concentration (ς_MNPs_) and volume (*V*_MF_) of MNPs. Conditions that satisfy the temperature increase Δ*T* > 6 °C lie above the white curve (which indicates the increase Δ*T* = 6 °C) (**a**) Comsol Multiphysics model for measuring temperature at the edge of the stimulation region; (**b**) *V*_MF_ = 0.6 µL; (**c**) *V*_MF_ = 1.2 µL; (**d**) *V*_MF_ = 1.8 µL; (**e**) *V*_MF_ = 2.4 µL; and (**f**) *V*_MF_ = 3.0 µL.

**Figure 5 ijms-20-02873-f005:**
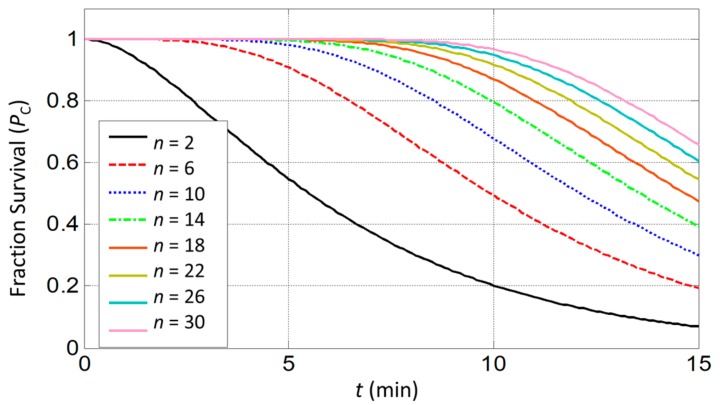
Estimation of cell survival rate in response to prolonged heat exposure at 44 °C for molecular targets *n* ranging from 2 to 30.

**Figure 6 ijms-20-02873-f006:**
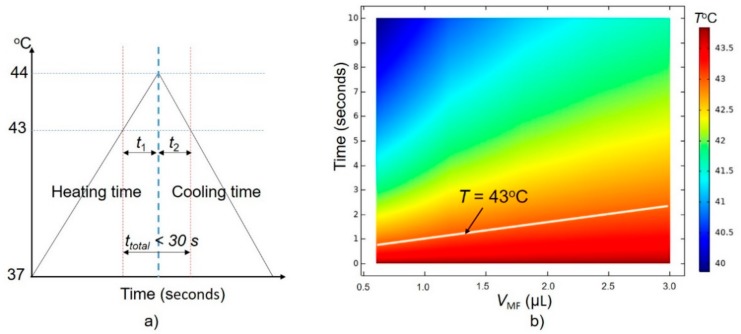
(**a**) A feasible condition for wireless magnetothermal simulation (WMS), (**b**) dependence of cooling time on *V*_MF_ with the initial temperature of the stimulation treatment equal to 44 °C.

**Figure 7 ijms-20-02873-f007:**
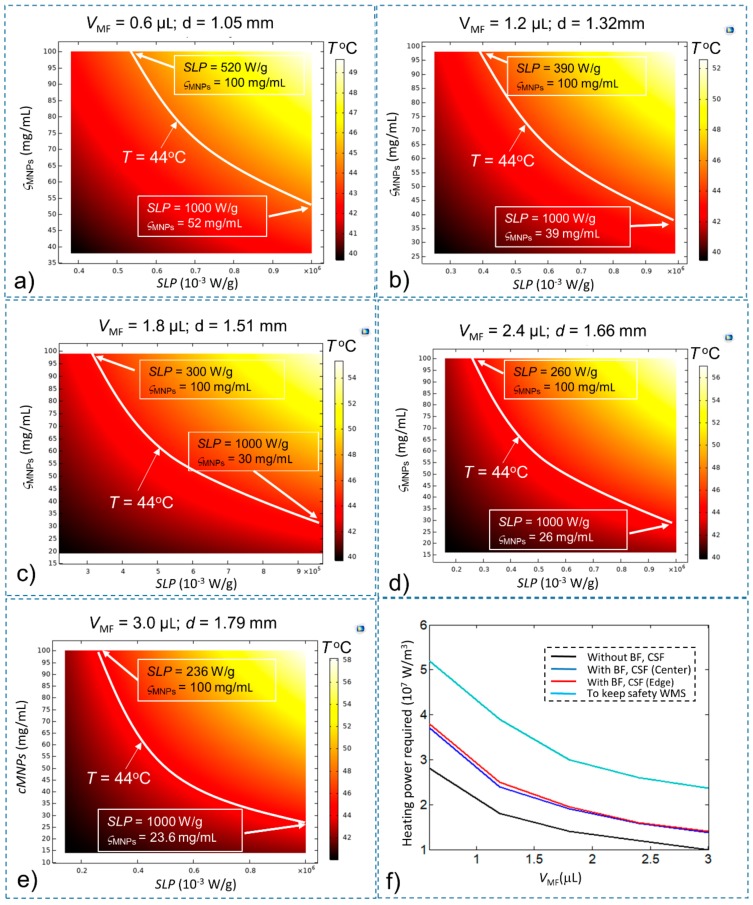
The temperature of the stimulation treatment region after time *t* = 25 s with T_initial_ = 43 °C for (**a**) *V*_MF_ = 0.6 µL; (**b**) *V*_MF_ = 1.2 µL; (**c**) *V*_MF_ = 1.8 µL; (**d**) *V*_MF_ = 2.4 µL; (**e**) *V*_MF_ = 3.0 µL; and (**f**) the minimum heating power *Q*_MNPs_ required for safe execution of WMS (compared with other conditions).

**Figure 8 ijms-20-02873-f008:**
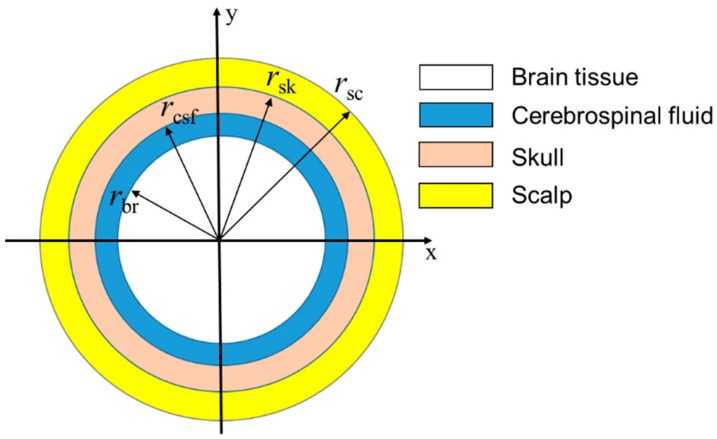
Simplified human brain model.

**Table 1 ijms-20-02873-t001:** The minimum amount of SLP, ς_MNPs_ and heating power *Q*_MNPs_ values required for WMS for different values of *V*_MF_ without considering blood flow (BF) and cerebrospinal fluid (CSF), determined using Fourier’s Law.

*V*_MF_ (µL)	Diameter of Stimulation Region *d* (mm)	Minimum SLP (W/g) at Maximum ς_MNPs_ (100 mg/mL)	Minimum ς_MNPs_ (mg/mL) at Maximum SLP(1000 W/g)	Maximum Δ*T* with Maximum SLP and ς_MNPs_ (°C)	Minimum Heating Power Required *Q*_MNPs_ (W/m^3^)
0.6	1.05	280	28	20	2.8 × 10^7^
1.2	1.32	180	18	30	1.8 × 10^7^
1.8	1.51	140	14	40	1.4 × 10^7^
2.4	1.66	120	12	50	1.2 × 10^7^
3.0	1.79	100	10	60	1.0 × 10^7^

**Table 2 ijms-20-02873-t002:** The minimum amount of SLP, ς_MNPs_ and heating power *Q*_MNPs_ values required for WMS at different values of *V*_MF_ with the consideration of BF and CSF from Figure 3 and Figure 4.

*V*_MF_ (µL)	Minimum SLP (W/g) at Maximum ς_MNPs_ (100 mg/mL)		Minimum ς_MNPs_ (mg/mL) at Maximum SLP (1000 W/g)		Maximum Δ*T* with Maximum SLP and ς_MNPs_ ( °C)		Minimum Heating Power Required *Q*_MNPs_ (W/m^3^)	
	Center	Edge	Center	Edge	Center	Edge	Center	Edge
0.6	370	380	37	38	16.31	15.94	3.7 × 10^7^	3.8 × 10^7^
1.2	240	250	24	25	25.18	24.58	2.4 × 10^7^	2.5 × 10^7^
1.8	190	195	19	19.5	32.36	31.57	1.9 × 10^7^	1.95 × 10^7^
2.4	158	160	15.8	16	38.60	37.65	1.58 × 10^7^	1.6 × 10^7^
3.0	138	140	13.8	14	44.23	43.13	1.38 × 10^7^	1.4 × 10^7^

**Table 3 ijms-20-02873-t003:** The minimum values of SLP, ς_MNPs_ and heating power *Q*_MNPs_ values for safe WMS execution with different *V*_MF_ values.

*V*_MF_ (µL)	Minimum SLP (W/g) at Maximum ς_MNPs_ (100 mg/mL)	Minimum ς_MNPs_ (mg/mL) at Maximum SLP (1000 W/g)	Minimum Heating Power Required *Q*_MNPs_ (W/m^3^)
0.6	520	52	5.2 × 10^7^
1.2	390	39	3.9 × 10^7^
1.8	300	30	3.0 × 10^7^
2.4	260	26	2.6 × 10^7^
3.0	236	23.6	2.36 × 10^7^

**Table 4 ijms-20-02873-t004:** Safe ranges of SLP, injection volume (*V*_MF_) and concentration (ς_MNPs_) values for investigation.

Name	Minimal Value	Maximal Value
SLP (W/g)	0	1000
*ς*_MNPs_ (mg/mL)	0	100
*V*_MF_ (µL)	0	3

**Table 5 ijms-20-02873-t005:** Physical parameters used in the simulation [13,20,46,47].

Parameters	Specific Heat *C* (J/(kg °C))	Density *ρ* (kg/m^3^)	Thermal Conductivity *k* (W/(m °C))	Blood Flow Rate *ω* (1/s)	Metabolic Rate *Q*_m_ (W/m^3^)	Radius (Adult) *r* (mm)	Temperature *T*_b_ (°C)	Initial Temperature *T*_0_ (°C)
Blood (b)	3800	1050	0.5	–	–	–	36.64	36.64
Scalp (sc)	4000	1000	0.34	0.00143	363	88	37.00	37.00
Skull (sk)	2300	1500	1.16	0.000143	70	87	37.00	37.00
CSF	3800	1007	0.61	0	0	84	37.00	37.00
Brain tissue (br)	3700	1050	0.51	0.008	10437	83	37.00	37.00
Brain surroundings	–	–	–	–	–	–	25.00	25.00

– is no value.

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
