# Peer review of "Theoretical Analysis for Wireless Magnetothermal Deep Brain Stimulation Using Commercial Nanoparticles"

_ijms, 2019, doi:10.3390/ijms20122873_

Round 1
Reviewer 1 Report
Deep brain stimulation (DBS) has been used in the treatment of various neurodegenerative diseases for many years. A wireless magnetothermal stimulation (WMS) is one of the brain stimulation methods that does not require implanted devices inside the brain. Stimulation of brain nerve cells equipped with temperature-sensitive ion channels (TRPV1) takes place via magnetic nanoparticles (MNPs), which are injected into the stimulation area. Temperature of 43–44oC opens the TRPV1 channels and activates brain neurons. The authors based on the Fourier law and bioheat equation examined the basic parameters of nanoparticles, namely: specific loss power (SLP), volume of stimulation (VMNPs), stimulation and cooling times (t1 and t2), and cytotoxic effects (Pc) to determine the real conditions of WMS stimulation. The simulations were carried out in a commercially available Comsol Multiphysics software.
In my opinion the obtained results are interesting and worth publishing in this journal.
The manuscript is written with proper English and good punctuation but standard of English may be improved.
Literature is mostly adequate but all references should be rewritten according to the International Journal of Molecular Science template style: https://www.mdpi.com/journal/ijms/instructions
In the references please use initials of authors names and remove the quotation marks “…”.
The most of cited literature are appropriate but I recommend referring some additional works from recent years, namely:
1) Nanoparticle-based and bioengineered probes and sensors to detect physiological and pathological biomarkers in neural cells, DOI: 10.3389/fnins.2015.00480
2) Electromagnetic Fields and Neurodegenerative Diseases, DOI: 10.15199/48.2019.01.33
3) Specifying the ferrofluid parameters important from the viewpoint of Magnetic Fluid Hyperthermia, DOI: 10.1109/WZEE.2015.7394040
4) Magnetophoretic Placement of Ferromagnetic Nanoparticles in RF Hyperthermia, DOI: 10.1109/PAEE.2017.8009003
In the manuscript, the authors did not give equation for Fourier law, from which they probably derived the equation (1).
Moreover, in the cited literature [15] before equation (1) I did not find such formulation. Please cite the paper where such formula is given or derive it. The authors did not explain what the SLP (or SAR) parameter in equation (1) means. It has not been defined in the paper. What formula was used for SLP calculation? The quantities: qMNPs in (1) and QMNPs in (6) are the same parameters? The quantities: cMNPs in (1) and C in (2) are the same parameters?
Please, add reference [15] for thermal conductivity k specified in line 178. The quantity k given in line 125 is the same parameter? Is not the S-parameter as indicated in line 248?. For condition Hf ≤ 5∙109 Am-1s-1 in line 181 add reference [16]. Also for quantities given in line 166 the reference is missing. Please give appropriate references for all physical parameters given in Table 2.
What is the 'wireless magnetothermal stimulation' described by the authors and how does this method differ from popular medical treatment called ‘transcranial magnetic brain stimulation’?
The authors did not say anything about the exciting magnetic field used in the simulation. What was the source of the AC magnetic field – which coils (turns, currents, frequencies) or plane waves are employed? Please, give the appropriate boundary conditions. What was the frequency and intensity of the magnetic field? A sinusoidal or pulsed field (as in the case of transcranial magnetic stimulation) was utilized? How they have been simulated in Comsol Multiphysics software?
In the abstract and introduction, the authors have written: 'A wireless magnetothermal stimulation (WMS) is ... implant free stimulation method'. I propose used here the term: 'implant devices free', because magnetic nanoparticles (MNPs) injected into the brain tissue can be treated as very small implants.
What the author mean by the term ‘internally produced MNPs’? I do not understand this.
The authors in the manuscript changed the order of the sections. In section 2 they describe: ‘Results and Discussion’ and in section 3: ‘Material and Method’. These sections should be swapped places as described in lines 60–61.
The work does not indicate what kind of magnetic nanoparticles were used in computer simulation? The authors should give the basic MNPs parameters including their size and other parameters necessary to determine SLP and power dissipated (p) in the sample ferrofluid. Why in the line 155, the authors have written: ‘we did not consider the material of the MNPs’? In the introduction, the authors have written about the superparamagnetic nanoparticles (SNPs): ‘superparamagnetic nanoparticles (SPs) with size range of 10–50 nm to dissipate heat via hysteretic power loss [9]’. Are there actually hysteresis heating in superparamagnetic particles of such sizes? What mechanisms cause the heating of these particles? In reference [9], authors write only about MNPs, term ‘superparamagnetic nanoparticles/properties’ does not appear at all. The authors also did not specify how (from which formulation) the specific loss power (SLP) and minimum heating power (p) of MNPs, given in Tables 3–5 were designated? This is very important and requires the explanation of the authors in the revised manuscript.
The quantity P given in line 124 is heating power of MNPs given in Tables or Pc given in equation (11)? Which power represent Figure 8f?
The clarity of drawings and graphs is generally good but Figures 2, 4, 5 and 8 should be enlarged. Descriptions in white colours are invisible. Figure 4 should be moved to section 2.3. When referring in the text, use the full term Figure X, Table Y, Equation Z. For better clarity before and after Figures and Tables should be added the empty lines. What are the radii of spheres given in Figure 1. They rather should be marked as rbr, rcsf, rsk, rsc.
Authors’ conclusions are poor and should be extended.
Moreover, there are some editorial mistakes in the manuscript text:
Please explain all abbreviations appear in the paper (abstract), e.g. SPs, MNPs.
Use a long dash ‘–’ when referring multiple citations and specifying numerical ranges, e.g. 43–44oC, [21–23].
Remove spaces between the number and temperature units, e.g. 44oC.
For other units add missing spaces, e.g. 25 s.
Remove dots at the end of individual subsections.
Remove the sign ‘x’ for the product in equations and in text change it into ‘∙’, e.g. 4∙107.
In the manuscript text please define all parameters appearing in the equations and tables, e.g. Qm.
All subscripts (which are not variables) should be written in non italic style, e.g. cMNPs, VMNPs.
All constants should be written in non italic style, e.g. π.
All variables should be written in italic style, e.g. SLP, VMNPs, ρ, n, Pc, ttotal.
Please use the spaces before and after the signs ‘≤’ and ‘=’, e.g. Hf ≤ 5.
In Tables 3 and 4 for temperature increase please use the symbol ΔT. Is the same temperature increase as indicated in line 84?
Please check all units appearing in the manuscript, e.g. W/(m2K), J/(kgoC), W/(moC).
Detailed remarks and suggested corrections are included in the attached pdf file with notes.
Summarizing, the paper contains some valuable conclusions of research nature especially in specifying the realizable guidelines for WMS stimulation, but it needs improvement before it will be accepted to print in this journal. In my opinion, the manuscript may deserves to be published in International Journal of Molecular Science after the major revision and author's corrections taking into account the recommendations indicated by the reviewers.

Author Response
The authors appreciate the constructive comments and suggestions from the reviewers. The manuscript has been revised based on these comments. The changes have been highlighted in the revised manuscript.
Point 1: About presentation errors:
- The clarity of drawings and graphs is generally good but Figures 2, 4, 5 and 8 should be enlarged. Descriptions in white colours are invisible. Figure 4 should be moved to section 2.3. When referring in the text, use the full term Figure X, Table Y, Equation Z. For better clarity before and after Figures and Tables should be added the empty lines. What are the radii of spheres given in Figure 1. They rather should be marked as rbr, rcsf, rsk, rsc.
- Use a long dash ‘–’ when referring multiple citations and specifying numerical ranges, e.g. 43–44oC, [21–23].
- Remove spaces between the number and temperature units, e.g. 44oC.
- For other units add missing spaces, e.g. 25 s.
- Remove dots at the end of individual subsections.
- Remove the sign ‘x’ for the product in equations and in text change it into ‘∙’, e.g. 4∙107.
- In the manuscript text please define all parameters appearing in the equations and tables, e.g. Qm.
- All subscripts (which are not variables) should be written in non italic style, e.g. cMNPs, VMNPs.
- All constants should be written in non italic style, e.g. π.
- All variables should be written in italic style, e.g. SLP, VMNPs, ρ, n, PC, ttotal.
- Please use the spaces before and after the signs ‘≤’ and ‘=’, e.g. Hf ≤ 5.
- Please check all units appearing in the manuscript, e.g. W/(m2K), J/(kgoC), W/(moC).
Response 1: All errors mentioned have been addressed in the revised manuscript. All Figures have been enlarged to increase clarity. All parameters appearing in the equations and tables have been properly defined. Please see the revised manuscript.
Point 2: The most of cited literature are appropriate but I recommend referring some additional works from recent years, namely:
1) Nanoparticle-based and bioengineered probes and sensors to detect physiological and pathological biomarkers in neural cells, DOI: 10.3389/fnins.2015.00480 (ok)
2) Electromagnetic Fields and Neurodegenerative Diseases, DOI: 10.15199/48.2019.01.33 (ok)
3) Specifying the ferrofluid parameters important from the viewpoint of Magnetic Fluid Hyperthermia, DOI: 10.1109/WZEE.2015.7394040 (chưa ok)
4) Magnetophoretic Placement of Ferromagnetic Nanoparticles in RF Hyperthermia, DOI: 10.1109/PAEE.2017.8009003
Response 2: Based on the reviewer’s comment, the specified references have been duly cited in the revised manuscript.
Point 3: Literature is mostly adequate but all references should be rewritten according to the International Journal of Molecular Science template style: https://www.mdpi.com/journal/ijms/instructions
In the references please use initials of authors names and remove the quotation marks “…”.
Response 3: Based on the reviewer’s comment, all references have been reformatted to conform to the journal style guidelines. Please see the revised manuscript.
Point 4: In the manuscript, the authors did not give equation for Fourier law, from which they probably derived the equation (1). Moreover, in the cited literature [15] before equation (1) I did not find such formulation. Please cite the paper where such formula is given or derive it. The authors did not explain what the SLP (or SAR) parameter in equation (1) means. It has not been defined in the paper. What formula was used for SLP calculation? The quantities: qMNPs in (1) and QMNPs in (6) are the same parameters? The quantities: cMNPs in (1) and C in (2) are the same parameters?
Response 4: Based on the reviewer’s comment, for better understanding, equation (1) of the original manuscript has been replaced by equations (1-4) in the revised manuscript, where equation (1) (new manuscript) is a formula from [28]. We have also included more details about SLP and a formula for calculation of measured SLP. The quantities: qMNPs in (1) and QMNPs in (6) are not the same, this has been corrected in the revised manuscript. The quantities: cMNPs in (1) (VMNPs in the revised manuscript) and C in (2) are also different, cMNPs (VMNPs in the revised manuscript) is concentration of MNPs whereas C is specific heat. To avoid confusion, we have changed cMNPs to VMNPs. Please see the revised manuscript.
Point 5: Please, add reference [15] for thermal conductivity k specified in line 178. The quantity k given in line 125 is the same parameter? Is not the S-parameter as indicated in line 248?. For condition H× f ≤ 5∙109 Am-1s-1 in line 181 add reference [16]. Also for quantities given in line 166 the reference is missing. Please give appropriate references for all physical parameters given in Table 2.
Response 5: The appropriate reference [29] for thermal conductivity k in Equation (1) has been added. The k given in line 125 (line 140 of the revised manuscript) is the S parameter mentioned in line 248 (line 294 of the revised manuscript), we have corrected line 140 of the revised manuscript accordingly. We have also added the reference [36] for the condition H× f ≤ 5∙109 Am-1s-1 in line 210. The references for all the physical parameters given in Table 2 (Table 5 in the revised manuscript) have been added in line 275. Please see the revised manuscript.
Point 6. What is the 'wireless magnetothermal stimulation' described by the authors and how does this method differ from popular medical treatment called ‘transcranial magnetic brain stimulation’?
Response 6: Transcranial magnetic stimulation (TMS) is a noninvasive form of brain stimulation in which a changing magnetic field is used to generate electric current in a specific area of the brain through electromagnetic induction. An electric pulse generator, or stimulator, is connected to a magnetic coil, which in turn is connected to the scalp. The stimulator generates a changing electric current within the coil that induces a magnetic field; this field then causes an induction of inverted electric charge within the brain itself. This technique, mentioned in [5] does not use magnetic nanoparticles. The 'wireless magnetothermal stimulation' also utilizes an alternating magnetic field (AMF), however, here the AMF is combined with magnetic nanoparticles (MNPs) of size range 10 – 50 nm to dissipate heat via hysteretic power loss. Both these techniques differ in terms of the mechanism by which they stimulate the brain. We have summarized this comment and included it in lines 41-43 of the revised manuscript.
Point 7: The authors did not say anything about the exciting magnetic field used in the simulation. What was the source of the AC magnetic field – which coils (turns, currents, frequencies) or plane waves are employed? Please, give the appropriate boundary conditions. What was the frequency and intensity of the magnetic field? A sinusoidal or pulsed field (as in the case of transcranial magnetic stimulation) was utilized? How they have been simulated in Comsol Multiphysics software?
Response 7: Based on the reviewer’s comment, the magnetic field strength and frequency values have been mentioned in the revised manuscript. We have used these magnetic field strength and frequency values to determine the range of SLP values while staying within the safety bounds, this is based on the references [33-36]. The magnetic field strength and frequency directly affect the SLP values as equation (2) in the revised manuscript. Therefore, using SLP values is sufficient to reflect the influence of the magnetic field strength and frequency. We have not used the magnetic field strength and frequency values in the calculations and simulations presented in this paper. However, the method used in this work to generate the AC magnetic field has been mentioned in the revised manuscript (line 210-217).
Point 8: In the abstract and introduction, the authors have written: 'A wireless magnetothermal stimulation (WMS) is ... implant free stimulation method'. I propose used here the term: 'implant devices free', because magnetic nanoparticles (MNPs) injected into the brain tissue can be treated as very small implants.
Response 8: We really appreciate the reviewer’s comment. Accordingly, we have changed ‘implant free’ to 'implant devices free'. Please see the revised manuscript (lines 11 and 44)
Point 9: What the author mean by the term ‘internally produced MNPs’? I do not understand this.
Response 9: Based on the reviewer’s comment, we have changed it to ‘with MNPs which are internally targeted into the brain’ and also included further explanation of this statement. Please see lines 12-13, 17-18, 46-47, and 58-59 of the revised manuscript.
Point 10: The authors in the manuscript changed the order of the sections. In section 2 they describe: ‘Results and Discussion’ and in section 3: ‘Material and Method’. These sections should be swapped places as described in lines 60–61.
Response 10: To follow the journal format, the Materials and Methods section has been moved to the section 3, and the Results and Discussion section has been moved to section 2. We have also updated the references and re-ordered the Figures, Equations, etc. Please see the revised manuscript.
Point 11: The work does not indicate what kind of magnetic nanoparticles were used in computer simulation? The authors should give the basic MNPs parameters including their size and other parameters necessary to determine SLP and power dissipated (p) in the sample ferrofluid. Why in the line 155, the authors have written: ‘we did not consider the material of the MNPs’? In the introduction, the authors have written about the superparamagnetic nanoparticles (SNPs): ‘superparamagnetic nanoparticles (SPs) with size range of 10–50 nm to dissipate heat via hysteretic power loss [9]’. Are there actually hysteresis heating in superparamagnetic particles of such sizes? What mechanisms cause the heating of these particles? In reference [9], authors write only about MNPs, term ‘superparamagnetic nanoparticles/properties’ does not appear at all. The authors also did not specify how (from which formulation) the specific loss power (SLP) and minimum heating power (p) of MNPs, given in Tables 3–5 were designated? This is very important and requires the explanation of the authors in the revised manuscript.
Response 11: Based on the reviewer’s comment, the MNPs selected to determine the ranges of SLP and VMNPs have been specified in Section 2 of the revised manuscript.
The parameters to determine SLP are given in the explanation of parameters for equation (3) (line 200 to 202). The power dissipated (QMNPs) in the sample ferrofluid can be determined by equation (2) (line 195), it can mainly be determined by SLP and VMNPs.
In accordance with the definition given in [9] (in Re [13] of the revised manuscript), the definition of WMS has been rewritten as: “….wireless magnetothermal stimulation (WMS) method has been suggested to provide a fast…. and magnetic nanoparticles (MNPs) with size range of 10 – 50 nm to dissipate hysteretic power loss.”
Since we have mentioned the material and size range of MNPs, the sentence ‘we did not consider the material of the MNPs’ in line 155 has been deleted. Please see lines 169-170 of the revised manuscript.
About the mechanisms involved in the heating of these particles, we have mentioned in the revised manuscript: “For particles having a size of micro or nano-scale, the heating due to the hysteresis loss is dominant. In the case of SNPs, these nano-sized particles exhibit a narrow hysteresis and the contributor to heat rise is the heating via Néelian and Brownian relaxation”. Please see the revised manuscript (lines 225 - 228).
Ranges of the specific loss power (SLP) values have been selected based on references [33-36]. Equation (3) for measurement of SLP has been included in the revised manuscript (line 200). The values of minimum heating power (QMNPs) of MNPs, given in Tables 3–5 (Table 1,2,3 in the revised manuscript), were selected according to equation (4) and the simulation done using the Comsol multiphysics software, all these values can increase the temperature by 6oC while satisfying the safety condition given in Figure 6. More explanation of this has been included in the revised manuscript (lines 72-73, 78, 122, 129, 159).
Please see the revised manuscript for all these changes.
Point 12: The quantity P given in line 124 is heating power of MNPs given in Tables or Pc given in equation (11)? Which power represent Figure 8f?
Response 12: The quantity P given in line 124 of the initial manuscript (line 139 of the revised manuscript) is PC given in equation (11) (equation (13) in the revised manuscript). Therefore, P in line 124 of the initial manuscript (line 139 of the revised manuscript) has been corrected to PC in line 139 of the revised manuscript. To avoid confusion, the heating power of MNPs given in the Tables has also been changed to QMNPs, please see Tables 1, 2 and 3 of the revised manuscript. The power represented in Figure 8f (Figure 7 in the revised manuscript) is QMNPs, we have mentioned this in the caption for Figure (7), please see line 148.
Point 13: In Tables 3 and 4 for temperature increase please use the symbol ΔT. Is the same temperature increase as indicated in line 84?
Response 13: Based on the reviewer’s comment, this error has been corrected in Tables 3 and 4 of the revised manuscript. This is the same temperature as indicated in line 84 of the initial manuscript (line 95 of the revised manuscript).
Point 14: Authors’ conclusions are poor and should be extended.
Response 14: Based on the reviewer’s comment, the conclusion section has been extended. Please see the revised manuscript.
Reviewer 2 Report
The manuscript reports on a theoretical analysis for wireless magnetothermal stimulation. The document is clearly written and can be followed with no major issues.
Comments are given below:
ABSTRACT
- The acronym "SPs" is not defined. Please do so.
- When discussing about the temperature, it is increased "up to 44oC" and not "up to less than 44oC".
INTRODUCTION
- It is not clear why implanted electrodes/chemicals plays a role in the penetration of visible light into tissues. Please explain.
- When dexcribing magnetic hyperthermia, a reltively recent review is available (Applied Physics Reviews 2, 041302 (2015)) and could be included in the reference list.
MATERIALS AND METHODS
- The section position is incorrect. Please switch with results and discussion.
- The authors indicate that Vmnp is "spherical microvolume of the ferrofluid". This quantity is not clearly explained: does it refer to the volume that the ferrofluid occupies or the particles? Please specify.
- Equation (1) in my understanding does not allow you to "quickly determine", but "to estimate".
- Why expression (11) is used? Which is its fundamental significance?
RESULTS AND DISCUSSIONS
- Figure 3 should have the x-axis scale modified so that the reader have a better view of the temperature differences among regions.
CONCLUSIONS
- I suggest the authors include in the conclusions section the conditions that optimally can make their study feasible to evaluated experimetally. In other words: what is the H*f threshold, optimum particle size and concentration in the ferrofluid, and so on so that the reader can have a prompt view of the simulated work significance.
Author Response
The authors appreciate the constructive comments and suggestions from the reviewers. The manuscript has been revised based on these comments. The changes have been highlighted in the revised manuscript.
Reviewer #2:
Point 1: The acronym "SPs" is not defined. Please do so.
Response 1: Based on the reviewer’s comment, the definition has been added. Please see the revised manuscript.
Point 2: When discussing about the temperature, it is increased "up to 44oC" and not "up to less than 44oC".
Response 2: This sentence has been revised. Please see line 14 of the revised manuscript.
Point 3: It is not clear why implanted electrodes/chemicals plays a role in the penetration of visible light into tissues. Please explain.
Response 3: In the initial manuscript, we wrote ‘... However, these methods often show un-specific stimulation and poor penetration of visible light into deep tissues, and sometimes require implanted devices for deep brain stimulation [2, 5].’ This may have caused the misunderstanding that implanted electrodes/chemicals play a role in the penetration of visible light into tissues. Therefore, we have change this sentence to ‘…However, these methods often show un-specific stimulation or poor penetration of visible light into deep tissues, or sometimes require implanted devices for deep brain stimulation’. Please see line 30 of the revised manuscript.
Point 4: When describing magnetic hyperthermia, a relatively recent review is available (Applied Physics Reviews 2, 041302 (2015)) and could be included in the reference list.
Response 4: The reference mentioned has been included in line 35 of the revised manuscript.
Point 5: The section position is incorrect. Please switch with results and discussion.
Response 5: In accordance with the journal format, we have changed the order of the sections. Please see the revised manuscript.
Point 6: The authors indicate that VMNPs is "spherical microvolume of the ferrofluid". This quantity is not clearly explained: does it refer to the volume that the ferrofluid occupies or the particles? Please specify.
Response 6: VMNPs has been changed to volume of magnetic fluid VMF (or injection volume) for clarity. Please see the revised manuscript.
Point 7: Equation (1) in my understanding does not allow you to "quickly determine", but "to estimate".
Response 7: We really appreciate reviewer’s comment. "quickly determine" has been changed to "to estimate". Please see line 237 of the revised manuscript.
Point 8: Why expression (11) is used? Which is its fundamental significance?
Response 8: Thanks for your question. Since it is important to determine the suitable parameters for WMS, we have used equation (13) (Equation 11 in the initial manuscript) to determine the safety condition. This equation decides the values of SLP, VMNPs, heating time, and cooling time that are suitable for WMS.
Point 9: Figure 3 should have the x-axis scale modified so that the reader have a better view of the temperature differences among regions.
Response 9: Figure 2 (Figure 3 in the initial version) has been revised for clarity. Please see the revised manuscript (line 92-93).
Point 10: I suggest the authors include in the conclusions section the conditions that optimally can make their study feasible to evaluated experimetally. In other words: what is the H*f threshold, optimum particle size and concentration in the ferrofluid, and so on so that the reader can have a prompt view of the simulated work significance.
Response 10: Based on the reviewer’s comment, the conclusions section has been rewritten. Please see the revised manuscript.
Round 2
Reviewer 1 Report
A wireless magnetothermal stimulation (WMS) is one of the brain stimulation methods that does not require implanted devices inside the brain.The authors based on the Fourier law and bioheat equation examined the basic parameters of nanoparticles, namely: specific loss power (SLP), volume of stimulation (VMNPs), stimulation and cooling times (t1 and t2), and cytotoxic effects (Pc) to determine the real conditions of WMS stimulation. The simulations were carried out in a commercially available Comsol Multiphysics software.
Most of the reviewer comments have been included in the revised manuscript.
The authors should clearly state in the revised manuscript that they did not simulate electromagnetic field (EMF) source, but only included the values of QMNPs for a given values of f and H of external EMF in the target region of human brain, which greatly simplifies the analysis of the analysed problem. The revised manuscript shows that QMNPs in the equations (1) and (6) is the same parameter, while in response for reviewer's comments, the authors wrote that these are different quantities. It should be clarified.
I strongly recommend changing the order of the sections 2 and 3. First the description of the modelling and then the obtained results should be described in the final manuscript. Some figures should be moved after the text, where they are referring. Fig. 5 should be reduced in size. Its fonts are too large compared to figure caption. Specified radii in Fig. 8 are hardly visible. It should be enlarged.
To improve the manuscript I recommend citing two additional papers, namely:
1) Passive Shielding of Magnetic Field in Transcranial Magnetic Stimulation–Outline of the Problem, DOI: 10.1109/ATEE.2019.87248622) Cooling effects inside water-cooled inductors for Magnetic Fluid Hyperthermia, DOI: 10.1109/PAEE.2017.8008997
Please delete the website address in reference [16].
In references of conference papers the DOI number should be included.
Additional remarks and suggested corrections are included in the attached pdf file with notes.
In my opinion, the revised manuscript deserves to be published in International Journal of Molecular Science after minor revision and author's corrections taking into account the recommendations indicated by the reviewers.

Author Response
The authors appreciate the constructive comments and suggestions from the reviewers. The manuscript has been revised based on these comments. The changes have been highlighted in the revised manuscript.
Reviewer #1:
Point 1: The authors should clearly state in the revised manuscript that they did not simulate electromagnetic field (EMF) source, but only included the values of QMNPs for a given values of f and H of external EMF in the target region of human brain, which greatly simplifies the analysis of the analysed problem. The revised manuscript shows that QMNPs in the equations (1) and (6) is the same parameter, while in response for reviewer's comments, the authors wrote that these are different quantities. It should be clarified.
Response 1: Thank you for your comment. To clearly state this matter, we have inserted additional text; “While operating within the safety conditions, magnetic field strength and frequency values can be represented by the SLP values, their relationship is shown in Equation (3). In addition, QMNPs depends on the SLP as shown in Equation (2). Thus, to simplify the analysis, we have used the SLP values instead of using both the magnetic field strength and frequency values to change QMNPs. Utilizing values of the SLP without creating a simulation model for the magnetic field can greatly simplify the simulation computation when it is applied to the brain model”, in the new revised manuscript. Also, equation (3) has been added to provide a clear relationship between the SLP and the magnetic field strength and frequency. Please see lines 239-244 and 208-213 of the new revised manuscript.
-The qMNPs only existed in equation (1) of the original manuscript and it was different from the QMNPs in equation (6) of the original manuscript. However, in the revised version, to avoid any misunderstanding, we have removed the qMNPs parameter and have only used QMNPs. So, in the current version, all mentions of QMNPs refer to the same parameter.
Point 2: I strongly recommend changing the order of the sections 2 and 3. First the description of the modelling and then the obtained results should be described in the final manuscript. Some figures should be moved after the text, where they are referring. Fig. 5 should be reduced in size. Its fonts are too large compared to figure caption. Specified radii in Fig. 8 are hardly visible. It should be enlarged.
Response 2: Thank you for your comment, in the original submitted manuscript the description of the modelling was given in section 2 and the obtained results were given in section 3. However, the Editor mentioned that according to the format of the journal section 2 should contain the Results and Discussion and the Materials and Methods should be in section 3. We do agree with the reviewer’s point of view, however we cannot change the order of sections due to the aforementioned journal guidelines.
-locations of all the figures have been changed based on the reviewer’s comment. The font size of Figure 5 has been reduced. The specified radii in Figure 8 have been modified to increase clarity. Please see the new revised manuscript.
Point 3: To improve the manuscript I recommend citing two additional papers, namely:
1) Passive Shielding of Magnetic Field in Transcranial Magnetic Stimulation–Outline of the Problem, DOI: 10.1109/ATEE.2019.8724862
2) Cooling effects inside water-cooled inductors for Magnetic Fluid Hyperthermia, DOI: 10.1109/PAEE.2017.8008997
Response 3: Based on the reviewer’s comment, the specified references have been duly cited in the new revised manuscript.
Point 4: Please delete the website address in reference [16].
In references of conference papers the DOI number should be included
Response 4: Based on the reviewer’s comment, we have deleted the website address in reference [16] (it is reference [17] in the new revised manuscript). In addition, DOI numbers have been added for all referred conference papers. Please see the new revised manuscript.
Reviewer 2 Report
The authors addressed the points indicated appropriately.
Author Response
The authors appreciate the constructive comments and suggestions from the reviewers.